# Uremia-Induced Gut Barrier Defect in 5/6 Nephrectomized Mice Is Worsened by *Candida* Administration through a Synergy of Uremic Toxin, Lipopolysaccharide, and (1➔3)-β-D-Glucan, but Is Attenuated by *Lacticaseibacillus rhamnosus* L34

**DOI:** 10.3390/ijms23052511

**Published:** 2022-02-24

**Authors:** Somkanya Tungsanga, Wimonrat Panpetch, Thansita Bhunyakarnjanarat, Kanyarat Udompornpitak, Pisut Katavetin, Wiwat Chancharoenthana, Piraya Chatthanathon, Naraporn Somboonna, Kriang Tungsanga, Somying Tumwasorn, Asada Leelahavanichkul

**Affiliations:** 1Department of Medicine, Division of Nephrology, Faculty of Medicine, Chulalongkorn University, Bangkok 10330, Thailand; s.tungsanga@gmail.com (S.T.); pkatavetin@yahoo.com (P.K.); kriangtungsanga@hotmail.com (K.T.); 2Department of Medicine, Division of General Internal Medicine-Nephrology, Faculty of Medicine, Chulalongkorn University, Bangkok 10330, Thailand; 3Department of Microbiology, Faculty of Medicine, Chulalongkorn University, Bangkok 10330, Thailand; mon-med@hotmail.com (W.P.); thansitadew@gmail.com (T.B.); jubjiibb@hotmail.com (K.U.); somying.tumwasorn@gmail.com (S.T.); 4Tropical Nephrology Research Unit, Department of Clinical Tropical Medicine, Faculty of Tropical Medicine, Mahidol University, Bangkok 10400, Thailand; wiwat.cha@mahidol.ac.th; 5Tropical Immunology and Translational Research Unit, Department of Clinical Tropical Medicine, Faculty of Tropical Medicine, Mahidol University, Bangkok 10400, Thailand; 6Department of Microbiology, Faculty of Science, Chulalongkorn University, Bangkok 10330, Thailand; memind01@gmail.com (P.C.); naraporn.s@chula.ac.th (N.S.); 7Microbiome Research Unit for Probiotics in Food and Cosmetics, Chulalongkorn University, Bangkok 10330, Thailand

**Keywords:** *Lacticaseibacillus rhamnosus*, *Candida*, gut-derived uremic toxins, gut leakage, 5/6 nephrectomy mice, chronic kidney disease

## Abstract

A chronic kidney disease (CKD) causes uremic toxin accumulation and gut dysbiosis, which further induces gut leakage and worsening CKD. Lipopolysaccharide (LPS) of Gram-negative bacteria and (1➔3)-β-D-glucan (BG) of fungi are the two most abundant gut microbial molecules. Due to limited data on the impact of intestinal fungi in CKD mouse models, the influences of gut fungi and *Lacticaseibacillus rhamnosus* L34 (L34) on CKD were investigated using oral *C. albicans*-administered 5/6 nephrectomy (5/6Nx) mice. At 16 weeks post-5/6Nx, *Candida*-5/6Nx mice demonstrated an increase in proteinuria, serum BG, serum cytokines (tumor necrotic factor-α; TNF-α and interleukin-6), alanine transaminase (ALT), and level of fecal dysbiosis (Proteobacteria on fecal microbiome) when compared to non-*Candida*-5/6Nx. However, serum creatinine, renal fibrosis, or gut barrier defect (FITC-dextran assay and endotoxemia) remained comparable between *Candida*- versus non-*Candida*-5/6Nx. The probiotics L34 attenuated several parameters in *Candida*-5/6Nx mice, including fecal dysbiosis (*Proteobacteria* and *Bacteroides*), gut leakage (fluorescein isothiocyanate (FITC)-dextran), gut-derived uremic toxin (trimethylamine-*N*-oxide; TMAO) and indoxyl sulfate; IS), cytokines, and ALT. In vitro, IS combined with LPS with or without BG enhanced the injury on Caco-2 enterocytes (transepithelial electrical resistance and FITC-dextran permeability) and bone marrow-derived macrophages (supernatant cytokines (TNF-α and interleukin-1 β; IL-1β) and inflammatory genes (*TNF-α*, *IL-1β*, *aryl hydrocarbon receptor*, and *nuclear factor-κB*)), compared with non-IS activation. These injuries were attenuated by the probiotics condition media. In conclusion, *Candida* administration worsens kidney damage in 5/6Nx mice through systemic inflammation, partly from gut dysbiosis-induced uremic toxins, which were attenuated by the probiotics. The additive effects on cell injury from uremic toxin (IS) and microbial molecules (LPS and BG) on enterocytes and macrophages might be an important underlying mechanism.

## 1. Introduction

Chronic kidney disease (CKD) has been recognized as an extensive worldwide burden for decades [1], causing an accumulation of various metabolic chemicals known as “uremic toxins”. Such toxins mainly derived from food components or metabolic activities in the body can contribute to various complications, such as cardiovascular diseases, pulmonary problems, and CKD progression [2]. Some circulating uremic toxins are formed in the GI tract, known as gut-derived uremic toxins [3], including trimethylamine-*N*-oxide (TMAO), indoxyl sulfate, p-cresol sulfate, hippuric acid, and phenylacetic acid [4]. Because of the defect of toxin elimination through kidneys in advanced CKD, the accumulated toxins are compensatorily excreted into the intestinal tract and selectively promote the overgrowth of pathogenic intestinal bacteria, so-called gut dysbiosis [5]. Dysbiosis enhances the production of gut-derived uremic toxins. The toxins (gut and non-gut derivatives) can impair intestinal epithelial tight junctions, which lead to translocation of microbial molecules from the gut into blood circulation, so-called gut leakage or gut translocation [6]. Although the intestine is a source of gut-derived uremic toxins, the toxins distribute throughout the body (including the intestine) induce damage to several cells, including enterocytes and renal tubules [7,8,9]. The vicious cycle that CKD causes uremic toxin accumulation and gut dysbiosis, which further induces gut leakage and worsens CKD, is referred to as the gut–kidney axis [10]. Gut translocation of microbial molecules and uremic toxins facilitates the inflammatory reaction and accelerate CKD progression [11].

Among microbial molecules in the gut, lipopolysaccharide (LPS) of Gram-negative bacteria and (1➔3)-β-D-glucan (BG) of fungi are the two most abundant molecules in the gut [10]. However, the impact of intestinal fungi in mouse models is underappreciated, as *Candida* albicans is less abundant in the mouse gut than in the human intestine [12]. The amount of *Candida* spp. in mouse feces is insufficient to be detectable in stool culture [12], which differs from the that of human feces [13]. Although gut fungi do not directly cause illness, they affect the gut microbiota and supply BG in the gut [14], contributing to the worsening of systemic inflammation following gut barrier defect (gut leakage). *Candida* administration in bilateral nephrectomy (acute kidney injury) mice induces more severe gut leakage and inflammatory responses [11]. However, there are still very few investigations on the impact of gut fungi on uremic disorders, and a CKD model with *C. albicans* presentation has never been explored. Because gut microbiota plays a major role in regulating the production of gut-derived uremic toxins and the toxins as well as endotoxemia worsen CKD progression [15,16], the use of probiotics might prevent gut dysbiosis, reduce the toxin, and delay CKD progression [17,18,19]. *Lacticaseibacillus rhamnosus* L34 (L34), a strain of intestinal flora isolated from the Asian population [20], improves gut permeability integrity in several animal models of acute illnesses [21]. Due to (i) the worsening renal fibrosis and CKD progression by systemic inflammation [22], (ii) systemic inflammation-induced gut leakage in acute uremia model [11], and (iii) the anti-inflammatory properties of probiotics [21,23,24], L34 administration might also help delay the CKD progression in the *Candida*-administered CKD mice.

Here, we explored the impact of *C. albicans* and L34 on renal histopathology, CKD progression, inflammatory markers, and gut leakage in the *Candida*-administered 5/6 nephrectomy (5/6Nx) mouse model. To understand the pathophysiologic effects of fungi on CKD, BG (the main fungal cell wall component) was used in vitro with LPS (a major component of Gram-negative bacterial cell wall) and indoxyl sulfate (a representative gut-derived uremic toxin) on enterocytes (Caco-2 cells) and macrophages (bone marrow-derived cells).

## 2. Results

### 2.1. Candida Administration Enhanced Proteinuria, Glucanemia, and Liver Damage in 5/6 Nephrectomy Mice That Were Attenuated by Lacticaseibacillus rhamnosus L34 (L34)

During 16 weeks of the observation, there was no mortality in 5/6Nx mice with or without *Candida* administration (data not shown). Chronic kidney disease (CKD) characteristics in 5/6Nx mice, compared with sham, was demonstrated by retardation of weight gain, increase in urine volume and proteinuria, increase in serum creatinine, presence of anemia, and gut barrier defect determined by endotoxemia, increased serum BG, and FITC-dextran assay, as early as 8 weeks after surgery (Figure 1A–H). In comparison with 5/6Nx mice without *Candida*, *Candida*-administered 5/6Nx mice had more prominent FITC-dextran (at 8 weeks), more proteinuria, and higher serum BG (glucanemia) (at 16 weeks) (Figure 1C,G,H). Among all of the time-point parameters, only proteinuria in *Candida*-5/6Nx mice demonstrated the worsening level at 16 weeks compared with 8 weeks (Figure 1C), supporting the CKD disease progression [25].

At 16 weeks post-5/6Nx, uremia induced (i) gut barrier defect, causing gut translocation of FITC-dextran (MW 4.4 kD), LPS (MW > 5 kDa), and BG (MW > 0.5 kDa) (Figure 1F–H) into the systemic circulation and (ii) gut-derived uremic toxins (trimethylamine *N*-oxide (TMAO) and indoxyl sulfate (IS)) and renal fibrosis which possibly induced systemic inflammation (TNF-α, IL-6, and IL-10), liver damage (alanine transaminase (ALT)), and proteinuria (Figure 1I–O and Figure 2). Although there was a trend of the more severe renal fibrosis in non-*Candida* 5/6Nx versus *Candida*-5/6Nx, the value was non-statistically significance (Figure 1O). Notably, the tubular dilatations in renal pathology of 5/6Nx model, despite a non-tubular ligation maneuver, might be due to the surgical scars (Figure 2). Among these parameters, the *Candida*-5/6Nx mice had more prominent gut leakage (FITC-dextran assay) (Figure 1H) that might contribute to higher proteinuria, serum cytokines (IL-6 and TNF-α) (Figure 1C), and liver injury (ALT) (Figure 1N) than the 5/6Nx mice, supporting systemic inflammation-induced proteinuria and hepatocyte injury [26,27]. Despite the alteration of some CKD characteristics after fungal gavage in *Candida*-5/6Nx mice, L34 administration in *Candida*-5/6Nx mice-attenuated proteinuria, gut leakage (FITC-dextran and glucanemia), serum levels of uremic toxins (TMAO and indoxyl sulfate), serum inflammatory cytokines, and liver damage (ALT) (Figure 1A–O).

### 2.2. Candida Altered Uremia-Induced Dysbiosis in 5/6 Nephrectomy Mice but Was Attenuated by Lacticaseibacillus rhamnosus L34

As gut *Candida* can induce dysbiosis in several models [28,29,30,31] and the gut microbiota can influence the gut-derived uremic toxins production [4], fecal microbiome analysis was examined. Although most of the fecal microbiota of 5/6Nx mice with and without *Candida* were comparable (Figure 3A–D), *Candida*-5/6Nx mice demonstrated more prominent *Proteobacteria* (pathogenic bacteria), *Helicobacter* spp. (uremic gastritis-associated bacteria) [32], and *Allobaculum* spp. (Gram-positive anaerobe with advantage property in obesity attenuation and disadvantage properties in mucus degradation [33,34]) (Figure 3C,D) without the difference in total abundance of Gram-negative bacteria (data not shown). The linear discriminant analysis effect size (LEfSe), the features that most likely explain the differences between groups [35], identified *Streptococcus* spp. (Gram-positive aerobes with uremic toxins association [36] and *Anaerovorax* spp. (Gram-positive anaerobic fermenters [37]) in *Candida*-5/6Nx mice (Figure 3E). In 5/6Nx without *Candida*, LEfSe indicated mostly Gram-negative anaerobic *Bacteroides* (*Prevotella* spp., *Alistipes* spp., and *Odoribacter* spp.), the dominant bacteria in the inflammatory intestine [38,39], with the limited *Firmicutes* bacteria (*Roseburia* spp. and *Clostridium* spp.), the dominant, primarily Gram-positive, anaerobes in the healthy condition [40] (Figure 3E). In parallel, the non-metric multidimensional scaling (NMDS), the pairwise dissimilarity between groups that the distance from axis representing the ranks of data [35], identified *Anaerovorax* spp. and *Mycoplasma* spp. as the representatives for *Candida*-5/6Nx and 5/6Nx, respectively, without the differences of microbial diversity (Chao analysis, estimating the richness, and Shannon analysis, evaluating the evenness) between groups (Figure 4A–C). These data implied *Candida*-induced gut dysbiosis in CKD mice.

In L34 administered *Candida*-5/6Nx, when compared with *Candida*-5/6Nx, *Bacteroides* (Gram-negative anaerobes with possible pathogenicity) [23], and *Proteobacteria* (pathogenic Gram-negative bacteria) were decreased, while *Cyanobacteria* (Gram-negative bacteria with toxins [41]) was increased in the phylum level analyses (Figure 3C). In the genus level analyses, L34 reduced *Helicobacter* spp., while increased *Clostridium* spp. and *Allobaculum* spp., compared with *Candida*-5/6Nx (Figure 3D). In parallel, LEfSe indicated mostly *Firmicutes* bacteria (the possible beneficial bacteria) [42], including *Clostridium* spp., *Butyricicoccus* spp., *Lactobacillus* spp., *Ruminococcus* spp., and *Oscillospira* spp., with an only Gram-negative bacteria; *Bacteroides acidifaciens* (obesity preventable bacteria) [43] (Figure 3E). Furthermore, NMDS identified *Marvinbryantia* spp., the cellulose-degrading *Firmicutes* bacteria [44], and *Odoribacter* spp., the butyrate (a beneficial short-chain fatty acid)-producing *Bacteroides* [45], in L34-administered *Candida*-5/6Nx mice without the alteration in microbial diversity (Figure 4A–C). The difference between L34-treated versus untreated mice indicating the possible benefits of probiotics in CKD supported the previous studies [46].

### 2.3. An Additive Effect of Indoxyl Sulfate, a Uremic Toxin, on Enterocytes and Macrophages Was Attenuated by Lacticaseibacillus Condition Media

Gut-derived uremic toxins frequently tested with indoxyl sulfate (IS) [47], can induce proinflammatory responses [48], which further enhance CKD progression [7,49]. With activation by a single molecule, the viability of enterocytes (Caco-2) reduced with 10 mM indoxyl sulfate or 400 μg/mL LPS (alone), but not BG alone (Figure 5A), implying the limited toxicity of BG on enterocytes. However, BG with IS (as low as 0.5 mM) reduced enterocyte viability (Figure 5A) supported an effect of the uremic toxin on enterocyte injury. Although the cell viability in the combined incubation of IS with LPS or LPS plus BG (LPS + BG) was comparable, such viability was higher than in the IS + BG combination (Figure 5A).

The effects of IS were also demonstrated by reduced transepithelial electrical resistance (TEER) and enhanced enterocyte permeability (FITC-dextran) when compared with the incubation by pathogen molecules alone (Figure 5B,C). The gut leakage was higher in the LPS + BG, compared to the activation by each molecule alone, and was highest in the LPS + BG with IS (Figure 5B,C). *Lacticaseibacillus* condition media (LCM) attenuated such injury from either LPS + BG or LPS + BG + IS; however, was higher than the control non-stimulated condition (Figure 5B,C). Of note, LPS at the lower concentrations (100 and 300 μg/mL) did not reduce the viability of Caco-2 cells and TEER (data not shown).

Because of the importance of macrophages in recognizing pathogen molecules [50,51], bone marrow-derived cells were used in the experiments. As such, IS alone (≥2 mM) or with LPS (or BG) reduced macrophage viability, and the combination between IS with LPS or BG did not reduce the toxic concentration of IS (Figure 6A). Although LPS + BG did not increase macrophage inflammation, indicated by supernatant cytokines (TNF-α and IL-1β), and the expression of inflammatory genes (*TNF-α*, *IL-1β*, *aryl hydrocarbon receptor*, and *NF-κB*), when compared to the LPS alone, which suggested the enhanced responses of IS against LPS, BG, and LPS + BG (Figure 6 and Figure 7). In comparison with LPS + BG, the *Lacticaseibacillus* condition media-treated group (LCM + LPS + BG) attenuated these parameters, although not reaching the level of the control groups (Figure 6 and Figure 7). These data supported the impact of gut fungi and uremic toxins on intestinal barrier defects and systemic inflammation.

## 3. Discussion

### 3.1. Gut Candida Altered Chronic Kidney Disease (CKD) in 5/6Nx Mice through Gut Dysbiosis, Intestinal Barrier Defect, and Systemic Inflammation

*C. albicans* in mouse feces is detectable only by PCR [52], not by culture [12], which differs from human conditions [13], *Candida* was orally administered in 5/6Nx mice to examine the impact of gut fungi in CKD. *Candida*-5/6Nx mice had more severe gut leakage (FITC-dextran at 8 weeks post-surgery) with higher serum BG (glucanemia) that possibly worsened proteinuria and liver damage (alanine transaminase (ALT)), and increased serum cytokines compared to 5/6Nx mice. Although *Candida* did not alter CKD severity determined by serum creatinine and kidney fibrosis, higher serum BG in *Candida*-5/6Nx mice enhanced responses against endotoxemia [7,53] that induced proteinuria and high ALT [26,27]. Although fungi were non-detectable in the feces of non-*Candida* 5/6Nx mice (data not shown), the detectable serum BG in these mice supported the role of BG from mouse chow in gut contents [54]. While *Candida* gavage in acute kidney injury (AKI) enhances gut leakage-induced inflammation and mortality [11], systemic inflammation from gut-*Candida* in CKD is not severe enough to increase the mortality partly due to the different compensation in acute versus chronic uremia [55]. An impact of acute uremia on enterocytes might be more potent than chronic uremia with a better adaptation to the cell micro-environments [55].

Uremia-induced gut dysbiosis is a result of an increase in intestinal excretion of the accumulated uremic toxins that promote the growth of gut pathogenic bacteria, increase gut-derived uremic toxins, and enhance systemic inflammation. These effects exert injury on renal vascular endothelium [56,57] and parenchymal cells [58,59] that causes a vicious cycle of uremic toxins induced gut dysbiosis, and the dysbiosis further enhanced CKD progression through the higher toxin accumulation. Then, this vicious cycle is possibly facilitated by gut fungi due to the enhanced serum BG with the presence of *Candida* in the gut. Interestingly, spontaneous glucanemia and endotoxemia (without systemic infection) in patients with CKD support the CKD-induced gut barrier defect [60] that might be correlated with gut dysbiosis. Indeed, gut fungi in 5/6Nx mice facilitated pathogenic Proteobacteria without alteration on Bacteroides and Firmicutes compared with non-*Candida* 5/6Nx. The gavage of *Candida* (live or heat-killed) in other mouse models facilitates fecal pathogenic bacteria [30,31,61], perhaps due to the BG fermentation properties of some bacteria [62]. Here, *Candida* also enhanced the growth of *Helicobacter* spp. and *Allobaculum* spp. that might be associated with uremic gastritis and mucus degradation [32,33]. Therefore, further exploration of gut fungi in CKD is interesting.

### 3.2. Lacticaseibacillus rhamnosus L34 (L34) Attenuated Candida-Administered 5/6Nx Mice through the Anti-Inflammatory Effect on Enterocytes and Macrophages

The attenuation of uremic enteropathy (gut dysbiosis and gut leakage) and gut-derived uremic toxins by L34 has been demonstrated [11]. However, the effect of L34 on the CKD model with *Candida* administration has never been explored. Despite a more profound inflammation in *Candida*-5/6Nx than non-*Candida* 5/6Nx, L34 attenuated disease severity in these mice, as indicated by renal injury (improve renal fibrosis and proteinuria, but not serum creatinine), gut barrier defect (FITC-dextran, but not LPS and BG), gut-derived uremic toxins (TMAO and IS), systemic inflammation (serum cytokines), and liver damage (ALT). Notably, serum creatinine has a limitation as a CKD biomarker [63] and has a higher molecular weight (MW) than 4.4 kDa of FITC-dextran. The MW of pathogen molecules (LPS and BG) [10] also varies. Therefore, after probiotic treatment, serum creatinine, LPS, and BG might be unchanged. In parallel, L34 improved gut dysbiosis as demonstrated by a reduction of the possible pathogenic bacteria (*Bacteroides*, *Proteobacteria*, and *Helicobacter* spp.) [11,32] with an increase in *Clostridium* spp. (the possible beneficial Fermenters) [64]. However, L34 increased *Cyanobacteria* (toxin-producing bacteria) [41] and *Allobaculum* spp. (bacteria with both advantage and disadvantage properties [32,33]). Although L34 did not alter *Firmicutes* (the high abundant bacteria in healthy condition) in the microbiome analysis (Figure 3A), LEfSe indicated several bacteria in the *Firmicutes* group in L34-treated mice (Figure 3E). As gut dysbiosis [5], gut leakage [6,7], gut-derived uremic toxins [6,65], and inflammatory cytokines are enhanced in advanced CKD, attenuation of these mechanisms might retard the CKD progression [11]. Our findings support such effects of probiotics on retardation of CKD progression.

In addition, the impact of indoxyl sulfate (IS), a representative gut-derived uremic toxin, on enterocytes and macrophages was evaluated in vitro using the extract of L34 in culture media. Indoxyl sulfate is a water-soluble form of indoxyl, a molecule converted from tryptophan amino acid by gut bacteria in the luminal side of the intestines [47] and is metabolized into IS by the liver and contacts the basolateral part of the enterocytes through blood circulation [9]. IS promotes the production of reactive oxygen species that directly induce cell damage and cell death in several organs [66,67]. Despite the resistance of Caco-2 against IS (reduced cell viability at 10 mM of IS), IS as low concentration as 0.5–1 mM could induce intestinal cell death [9] and enterocyte permeability (TEER and FITC-dextran assay) when combined with LPS or BG. These results imply an impact of IS on enterocytes during CKD-induced uremic intestine [68]. In parallel, macrophages were more susceptible to IS as the cell viability reduced in IS at 2 mM, but the combination of IS with LPS or BG did not decrease the IS concentration. Without IS, BG alone did not induce macrophage responses, and BG with LPS demonstrated the tendency of the additive proinflammatory effect but was not different from LPS activation alone. With IS, the additive proinflammatory effect was increased as IS + LPS + BG induced a higher level of inflammatory cytokines than IS + LPS. The additive effect is possibly due to the similar downstream signaling through the NFκB transcription factor of TLR-4 and Dectin-1, the pattern recognition receptors of LPS and BG, respectively, and aryl hydrocarbon receptor (the cytosolic receptor of IS) [69] as demonstrated in the working hypothesis figure (Figure 8). Despite the profound inflammation from combined IS with LPS and BG, LCM attenuated both enterocyte permeability defect and macrophage inflammation. These findings support the benefits of probiotics on CKD [70], possibly through anti-inflammatory exopolysaccharide [71]. Although the MW of IS, at 0.23 kDa, is small enough to pass through the normal intestinal tight junction (MW lower than 0.6 kDa) [10,72], several probiotics decrease serum IS [73,74], implying a decrease in bacteria with IS production property during advanced CKD. Probiotic tests on other CKD models might be interesting because of the possible difference in interference of gut dysbiosis and inflammatory responses in individual CKD models. For example, gut dysbiosis might be more prominent in the oral adenine-induced CKD model due to its direct effect on intestinal microbiota with inflammasome-related inflammatory mechanisms (crystal-mediated inflammation) [75,76,77,78]. With gut dysbiosis attenuation and the anti-inflammatory properties of the probiotics, further exploration in other CKD models and clinical studies for a potential application in CKD are warranted.

## 4. Materials and Methods

### 4.1. Animals

Animal care and experiments were performed according to the National Institutes of Health (NIH) criteria [79] using the approved protocol by the Institutional Animal Care and Use Committee of the Faculty of Medicine, Chulalongkorn University, Bangkok, Thailand (CU-ACUP No. 018/2562). Eight-week-old male C57BL/6 mice were purchased from Nomura Siam (Pathumwan, Bangkok, Thailand).

#### 4.1.1. *Candida*-Administered 5/6 Nephrectomy Model and the Probiotics

The 5/6 nephrectomy (5/6 Nx) surgery was performed via flank approach under isoflurane anesthesia. The upper and lower poles of the left kidney were removed, and a right nephrectomy was performed one week later. Only mice with a weight of the removed fragments from the left kidney to the right kidney weight in a ratio of 0.55–0.72 were included to ascertain that removal of the left kidney mass is optimum for the CKD development [25]. Then, the 5/6Nx mice were divided into 3 groups, including the phosphate buffer solution (PBS) control (5/6Nx PBS), *Candida*-administered 5/6Nx (5/6Nx + *Candida*), and 5/6Nx + *Candida* with the probiotic administration (5/6Nx + *Candida* + L34). As such, *Candida albicans* from the American Type Culture Collection (ATCC 90028) (Fisher Scientific, Waltham, MA, USA) were cultured overnight on Sabouraud dextrose broth (SDB) (Oxoid, Hampshire, UK) at 35 °C for 48 h before enumeration using a hemocytometer. *C. albicans* at 1 × 10^6^ CFU in a 0.5 mL PBS or PBS alone was orally administered on alternate days starting from 4 weeks after the right kidney removal of the 5/6Nx procedure. For the probiotics, *Lacticaseibacillus rhamnosus* L34 (L34) from the stock (Chulalongkorn University) was cultured on deMan-Rogosa-Sharpe (MRS) agar (Oxoid™, Hampshire, UK) under anaerobic conditions (10% CO_2_, 10% H_2_, and 80% N_2_) using gas generation sachets (AnaeroPack^®^-Anaero; Mitsubishi Gas Chemical, Tokyo, Japan) at 37 °C for 48 h before use. After 4 weeks of the right nephrectomy, L34 at 1 × 10^7^ colony-forming unit (CFU) in 0.25 mL PBS were orally administered 3 times a week (on the different days of *C. albicans* gavage) until 16 weeks post-5/6Nx. Another group of mice was the sham operation to identify renal vessels before abdominal closure (Sham group). The L34-administered 5/6Nx mice without *Candida* have not been demonstrated here because the benefits of L34 on non-*Candida*-administered mice with uremia are previously published [11].

#### 4.1.2. Mouse Sample Analysis

Blood and urine parameters at 0 and 8 weeks of 5/6Nx were determined from blood collection (tail nicking) and urine collection (using the metabolic cage) (Hatteras Instruments, Cary, NC, USA) at 3 days before and 8 weeks after right kidney removal, respectively. Hematocrit was measured by the microhematocrit method with the Coulter Counter (Hitachi 917; Boehringer Mannheim GmbH, Mannheim, Germany). Serum creatinine and 24-h albuminuria were measured by colorimetric method (QuantiChrom™ Creatinine Assay Kit, BioAssay System, Hayward, CA, USA) and enzyme-linked immunosorbent assay (ELISA) (Albuwell M, Exocell™, Philadelphia, PA, USA), respectively. Serum TMAO was determined by liquid chromatography–mass spectrometry (LC-MS/MS) using a silica column (Luna^®^ silica; 00G-4274-E0, Phenomenex^®^, Torrance, CA, USA) [80]. Various concentrations of non-isotopically labeled TMAO standards were spiked into the control serum to construct calibration curves. The internal standard d9-TMAO was used for quantification and calculating the recovery rate of TMAO. Serum indoxyl sulfate was determined by high-performance liquid chromatography (HPLC Alliance^®^ 2695; Waters, Zellik, Belgium) [81]. Serum cytokines (TNF-α, IL-6, and IL-10) and liver damage were evaluated by ELISA (Invitrogen, Carlsbad, CA, USA) and EnzyChrom Alanine Transaminase assay (EALT-100, BioAssay, Hayward, CA, USA), respectively. For renal histopathologic studies, the remaining kidneys were fixed in 10% formalin, paraffin-embedded, and stained with Masson’s trichrome and Hematoxylin and Eosin (H&E) colors [22]. The area of renal fibrosis in Masson’s trichrome stained sections was determined by the computerized image analysis software (ImageJ © software, Bethesda, MD, USA) in a 200× magnification field with 10 fields per sample.

#### 4.1.3. Gut Permeability Determination

Gut permeability was determined by fluorescein isothiocyanate dextran (FITC-dextran) assay and the spontaneous elevation of LPS as well as BG in blood [23,82,83,84]. The detection of FITC-dextran (an intestinal nonabsorbable molecule) in serum at 3 h after an oral administration or spontaneous serum elevation of endotoxin without systemic inflammation indicates gut permeability defect (gut leakage) [23,82,83,84]. In the FITC-dextran assay, 12.5 mg of FITC-dextran (4.4 kDa) (FD4; Sigma-Aldrich^®^, St. Louis, MO, USA) was orally administered. At 3 h thereafter, FITC-dextran in serum was measured from blood samples by Fluorospectrometer (NanoDrop™ 3300; Thermo Fisher Scientific™, Wilmington, DE, USA). Serum LPS (endotoxin) and BG was evaluated by HEK-Blue LPS Detection Kit 2 (InvivoGen™, San Diego, CA, USA) and Fungitell^®^ assay (Associates of Cape Cod, Falmouth, MA, USA), respectively. Due to the lower limit of the standard curve of the test, the value less than 0.01 EU/mL and 7.8 pg/mL in LPS and BG assay, respectively, were recorded as 0.

#### 4.1.4. Fecal Microbiome Analysis

Feces (0.25 g per mouse) were collected from the mice in different cages per group to avoid the influence of allocoprophagy (a habit of ingesting the feces from other mice) and fecal microbiome analysis was performed as previously described [39]. The metagenomic DNA was extracted from the prepared samples using a DNAeasy Kit (Qiagen, Hilden, Germany) with DNA quality assessment using Nanodrop spectrophotometry. Universal prokaryotic primers 515F (5′-GTGCCAGCMGCCGCGGTAA-3′) and 806R (5′-GGACTACHVGGGT WTCTAAT-3′) with appended 50 Illumina adapters and 30 Golay barcode sequences were used for 16S rRNA gene V4 library construction.

### 4.2. In Vitro Experiments

To test a possible effect of BG (a major *Candida* component) on intestine and systemic inflammation, indoxyl sulfate was used as a representative gut-derived uremic toxin on enterocytes (Caco-2) (American Type Culture Collection, Manassas, VA, USA) and bone marrow-derived macrophages. Macrophages were derived from the bone marrow of 8 week-old mice using the modified Dulbecco’s Modified Eagle’s Medium (DMEM) and conditioned media of the L929 cell line (a source of macrophage colony-stimulating factor) as previously mentioned [85]. The Caco-2 (at 2 × 10^6^ cells/well) or bone marrow-derived macrophages (BMDM) (at 5 × 10^4^ cells/well) in DMEM were incubated with the different concentrations of indoxyl sulfate (Sigma-Aldrich) alone or together with BG (CM-Pachyman) (Megazyme, Bray, Ireland) at 100 μg/mL with or without LPS (*Escherichia coli* O26:B6) (Sigma-Aldrich) at 400 μg/mL (for Caco-2 cells) and 100 μg/mL (for macrophages) before the determination of cell viability with tetrazolium dye 3-(4,5-dimethylthiazol-2-yl)-2,5-diphenyltetrazolium (MTT) solution [86]. The 24 h-activated cells were incubated with 0.5 mg/mL of MTT solution (Thermo Fisher Scientific) for 2 h at 37 °C in the dark, and MTT was removed and diluted with dimethyl sulfoxide (DMSO; Thermo Fisher Scientific) before measurement with a Varioskan Flash microplate reader at an absorbance of optical density at 570 nm. The different concentrations of LPS in Caco-2 and macrophages are due to the variable LPS resistance between these cells [87].

Additionally, the 24 h stimulated Caco-2 cells were used to evaluate the integrity of intestinal cells as determined by transepithelial electrical resistance (TEER) [88]. To establish the confluent monolayer, Caco-2 cells at 5 × 10^4^ cells per well were seeded onto the upper compartment of 24-well Boyden chamber trans-well, using DMEM-high glucose supplemented with 20% Fetal Bovine Serum (FBS), 1% HEPES, 1% sodium pyruvate, and 1.3% Penicillin/Streptomycin for 15 days. TEER, in ohm (Ω) × cm^2^, was measured with an epithelial volt-ohm meter (EVOM2™, World precision instruments, Sarasota, FL, USA) by placing electrodes in the supernatant at the basolateral chamber and in the apical chamber. The TEER values in media culture without Caco-2 cells were used as a blank and were subtracted from all other measurements. In parallel, the enterocyte permeability was also evaluated by the FITC-dextran test following the published protocols [89]. As such, 5 μL of FITC-dextran (4.4 kDa) (Sigma-Aldrich) at 10 mg/mL was added to the apical side of the trans-well chamber with 24 h stimulated Caco-2 cells, and the FITC-dextran from the basolateral side of the trans-well plate was measured at 1 h after incubation using a Fluorospectrometer (NanoDrop 3300) (ThermoFisher Scientific). The concentration of FITC-dextran from the basolateral side, in percentage relative to the dose on the apical side, represents the severity of permeability defect of Caco-2 cells.

In macrophages, the supernatant cytokines (TNF-α and IL-1β) in 24 h stimulated macrophages (at 5 × 10^4^ cells/well) were evaluated by ELISA (Invitrogen, Carlsbad, CA, USA). Furthermore, the expression of selected genes related to inflammatory responses was examined by quantitative reverse transcription-polymerase chain reaction (qRT-PCR) in relative to β-actin (a housekeeping gene) with the 2^−ΔΔCT^ method using cDNA (SuperScript™ Vilo™ cDNA synthesis assay, Invitrogen™, Waltham, MA, USA) prepared from 50 ng of TRIzol-extracted total RNA (invitrogen™, Waltham, MA, USA) by a qPCR machine (LightCycler^®^ 2.0, Roche Diagnostics, Indianapolis, IN, USA). The primers for qRT-PCR were as follows; Tumor necrosis factor α (TNF-α) forward 5′-CCTCACACTCAGATCATCTTCTC-3′ and reverse 5′-AGATCCATGCCGTTGGCCAG-3′; Interleukin-1β (IL-1β) forward 5′-GAAATGCCACCTTTTGACAGTG-3′ and reverse 5′-TGGATGCTCTCATCAGGACAG-3′; Aryl hydrocarbon receptor (Ahr) forward 5′-ACCACTTAGAGCACCACTA-3′ and reverse 5′-AGAACTTCAATCAGACATACACAA-3′ and Nuclear factor-κB (NF-κB) forward 5′-CTTCCTCAGCCATGGTACCTCT-3′ and reverse 5′-CAAGTCTTCATCAGCATCAAACTG-3′.

### 4.3. Statistical Analysis

Analyzed data are presented as mean ± standard error (SE) using GraphPad Prism version 9.0 software (La Jolla, CA, USA). Statistical significance was determined by one-way analysis of variance (ANOVA) followed by Tukey’s analysis or Student’s *t*-test for comparisons among groups and between the 2 groups, respectively. The time-point experiments were analyzed by the repeated measures ANOVA. All statistical analyses were performed with Stata 16.0 software (StataCorp, TX, USA) and Graph Pad Prism version 7.0 software (La Jolla, CA, USA). The *p*-value of <0.05 was considered statistically significant.

## 5. Conclusions

*Candida* administration enhanced leaky gut and inflammatory responses in 5/6Nx CKD mice through the additive inflammatory activation of IS with LPS and BG. *L. rhamnosus* L34 attenuated the severity of *Candida*-5/6Nx mice, partly through the improved enterocyte integrity and the induced anti-inflammatory macrophages. Some probiotics would be an important adjuvant therapy in patients with CKD in the near future.

## Figures and Tables

**Figure 1 ijms-23-02511-f001:**
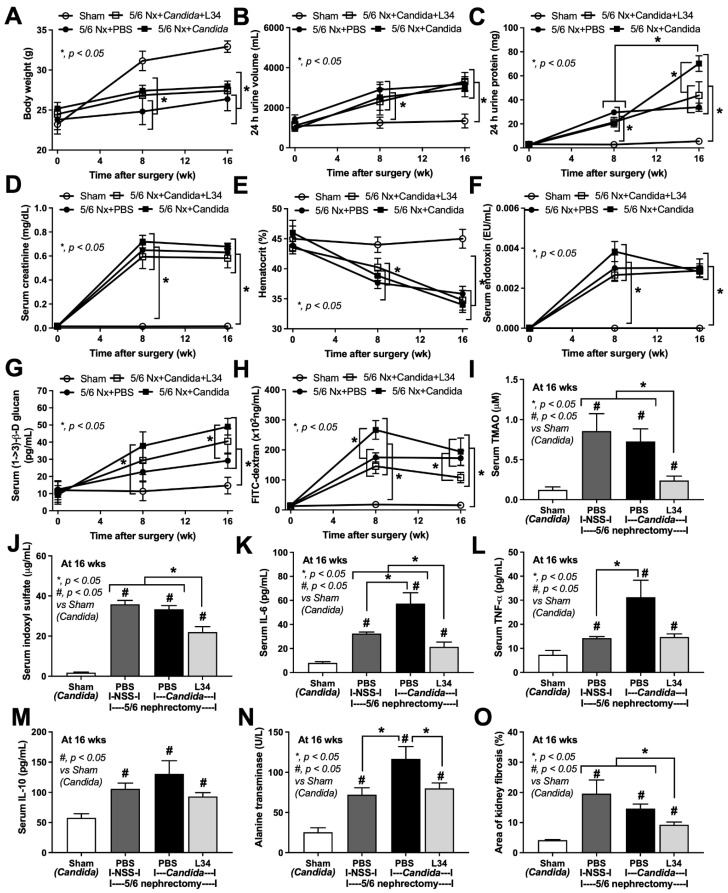
The characteristics of Sham control, 5/6 nephrectomy mice with phosphate buffer solution (5/6 Nx + PBS), *Candida*-administered 5/6 nephrectomy mice with *Lacticaseibacillus rhamnosus* L34 (5/6 Nx + *Candida* + L34) or without the probiotics (5/6 Nx + *Candida*) as indicated by the time-point of: (**A**) body weight; (**B**) 24-h urine volume; (**C**) 24-h urine protein; (**D**) serum creatinine; (**E**) hematocrit, and gut barrier defect: (**F**) endotoxemia; (**G**) serum (1➔3)-β-D-glucan; and (**H**) FITC-dextran; with the parameters at 16 weeks of the model, including serum gut-derived uremic toxins: (**I**) Trimethylamine N-oxide (TMAO) and (**J**) indoxyl sulfate; serum cytokines: (**K**) IL-6; (**L**) TNF-α; and (**M**) IL-10; (**N**) liver enzyme (alanine transaminase); and (**O**) renal fibrosis score are demonstrated (n = 6–10/time-point or group).

**Figure 2 ijms-23-02511-f002:**
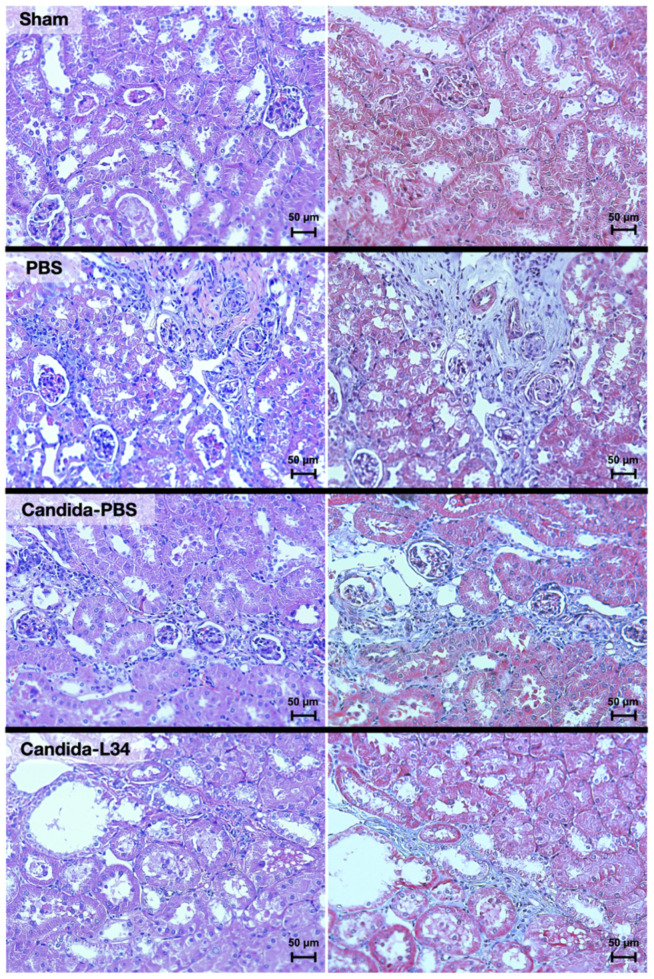
The representative kidney histopathological pictures stained by Hematoxylin and Eosin (H&E) (**left**) or Masson’s Trichrome color (**right**) (original magnification 200×) of Sham control, 5/6 nephrectomy mice with phosphate buffer solution (5/6Nx + PBS), *Candida*-administered 5/6 nephrectomy mice with *Lacticaseibacillus rhamnosus* L34 (5/6 Nx + *Candida* + L34) or without the probiotics (5/6 Nx + *Candida*) are demonstrated.

**Figure 3 ijms-23-02511-f003:**
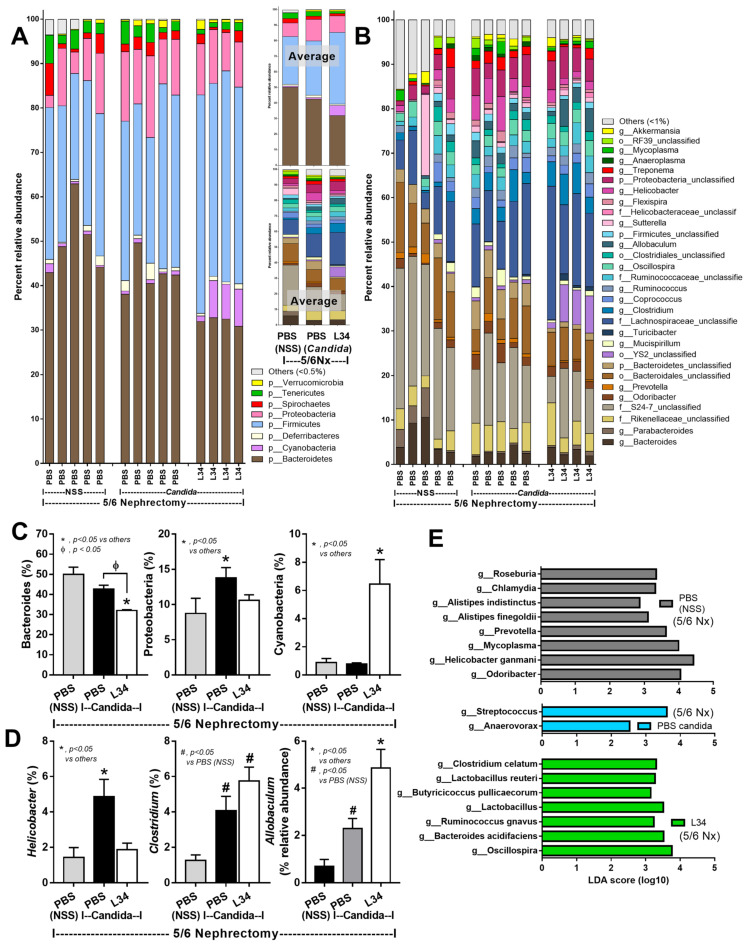
The fecal microbiome analysis 5/6 nephrectomy mice with phosphate buffer solution (5/6Nx + PBS), *Candida*-administered 5/6 nephrectomy mice with *Lacticaseibacillus rhamnosus* L34 (5/6 Nx + *Candida* + L34) or with normal saline solution (NSS) control (5/6 Nx + *Candida* + NSS) as indicated by the relative abundance in (**A**) phylum and (**B**) genus; with the average relative abundance in phylum and genus; (**C**,**D**) the graph presentation of relative abundance in phylum and genus levels; and (**E**) The linear discriminant analysis effect Size (LEfSe) are demonstrated.

**Figure 4 ijms-23-02511-f004:**
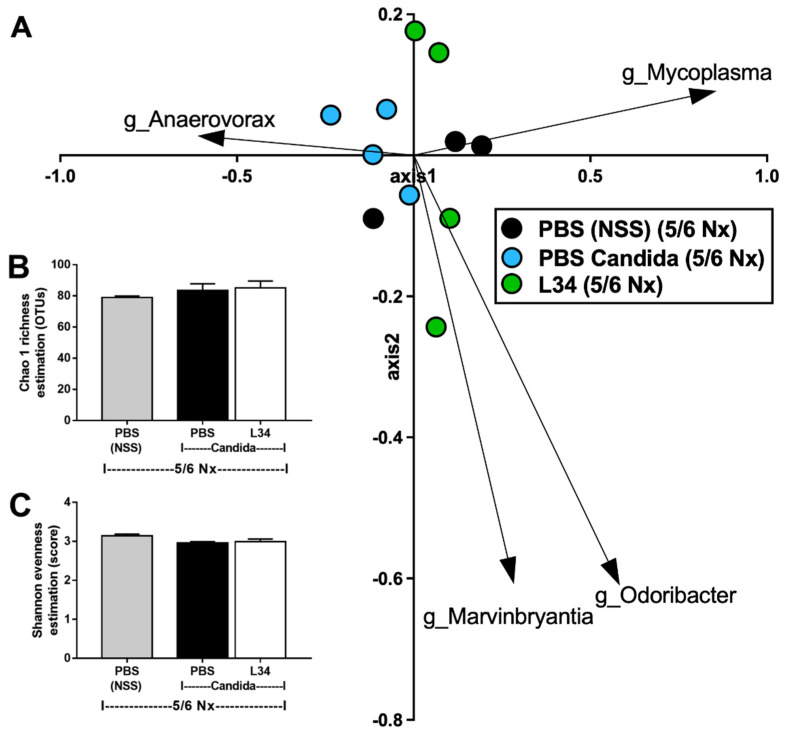
The fecal microbiome analysis 5/6 nephrectomy mice with phosphate buffer solution (5/6 Nx + PBS), *Candida*-administered 5/6 nephrectomy mice with *Lacticaseibacillus rhamnosus* L34 (5/6 Nx + *Candida* + L34) or with normal saline solution (NSS) control (5/6 Nx + *Candida* + NSS) as indicated by (**A**) the non-metric multidimensional scaling (NMDS); the alpha diversity: (**B**) microbial diversity of Chao and (**C**) Shannon estimation are demonstrated.

**Figure 5 ijms-23-02511-f005:**
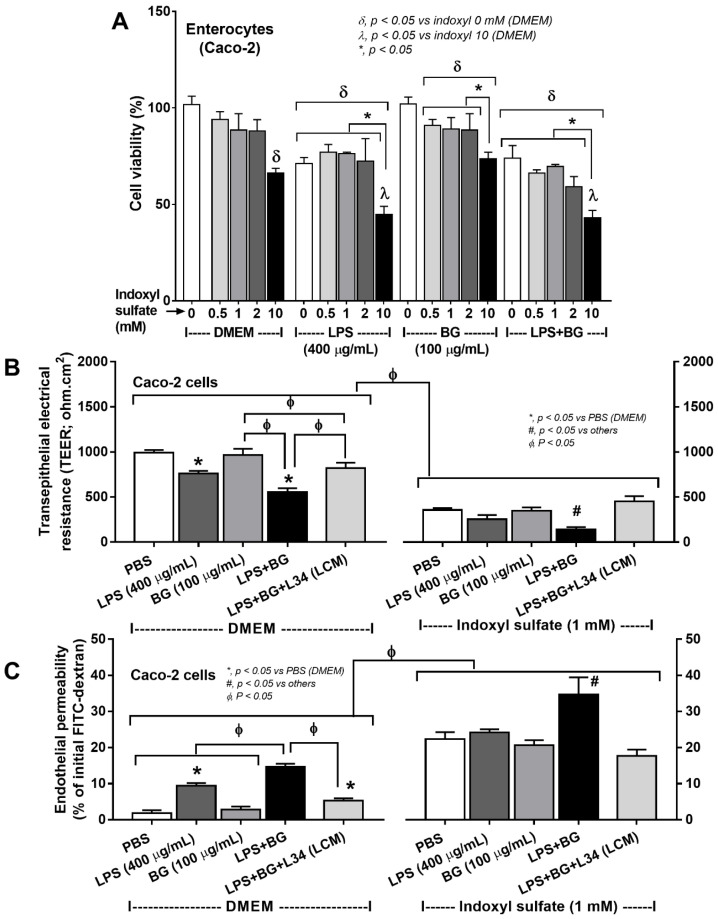
(**A**) The cell viability (MTT assay) of enterocytes (Caco-2 cells) after incubation by the different concentrations of indoxyl sulfate together with culture media control (using DMEM; Control) or lipopolysaccharide (LPS) at 400 μg/mL with or without (1➔3)-β-D-glucan (BG) at 100 μg/mL are demonstrated. (**B**) The transepithelial electrical resistance (TEER) and (**C**) enterocyte permeability (FITC-dextran 4.4 kDa) in stimulated Caco-2 cells with or without 1 mM indoxyl sulfate, a representative uremic toxin, are also demonstrated (independent triplicate experiments were performed for the in vitro experiments).

**Figure 6 ijms-23-02511-f006:**
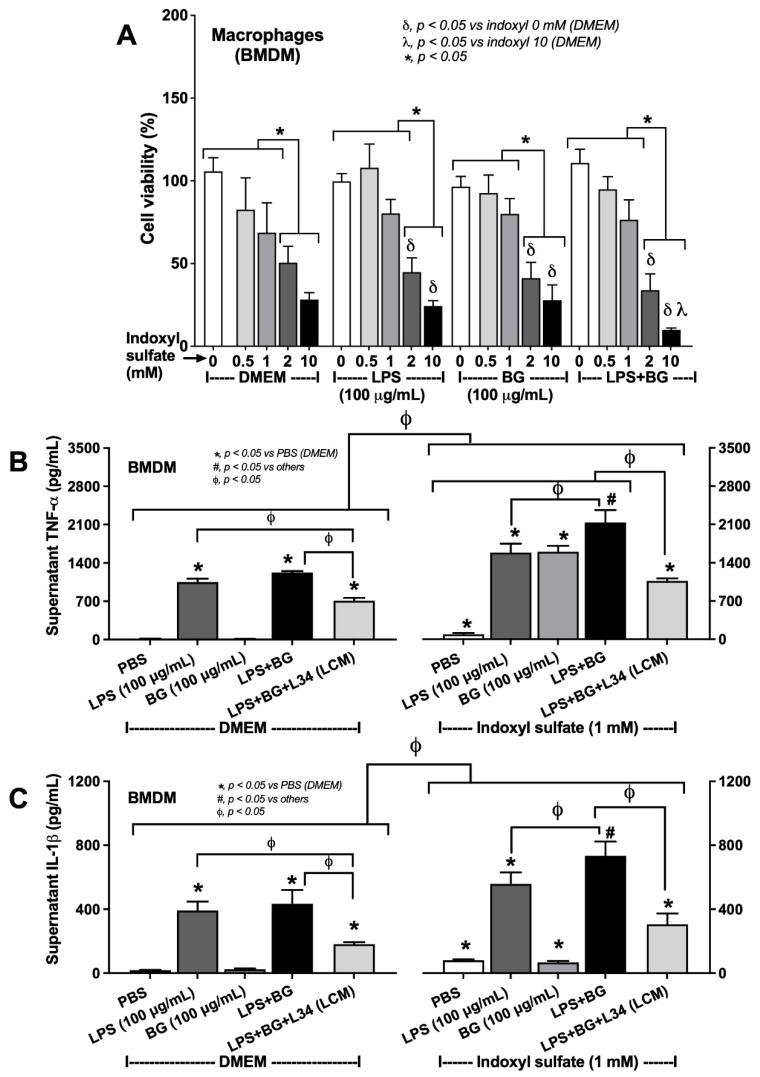
(**A**) The cell viability (MTT assay) of bone marrow-derived macrophages (BMDM) after incubation by the different concentrations of indoxyl sulfate together with culture media control (using DMEM) or lipopolysaccharide (LPS) at 100 μg/mL with or without (1➔3)-β-D-glucan (BG) at 100 μg/mL are demonstrated. The supernatant cytokines: (**B**) TNF-α; and (**C**) IL-1β in stimulated BMDM with or without 1 mM indoxyl sulfate, a representative uremic toxin, is also demonstrated (independent triplicate experiments were performed for the in vitro experiments).

**Figure 7 ijms-23-02511-f007:**
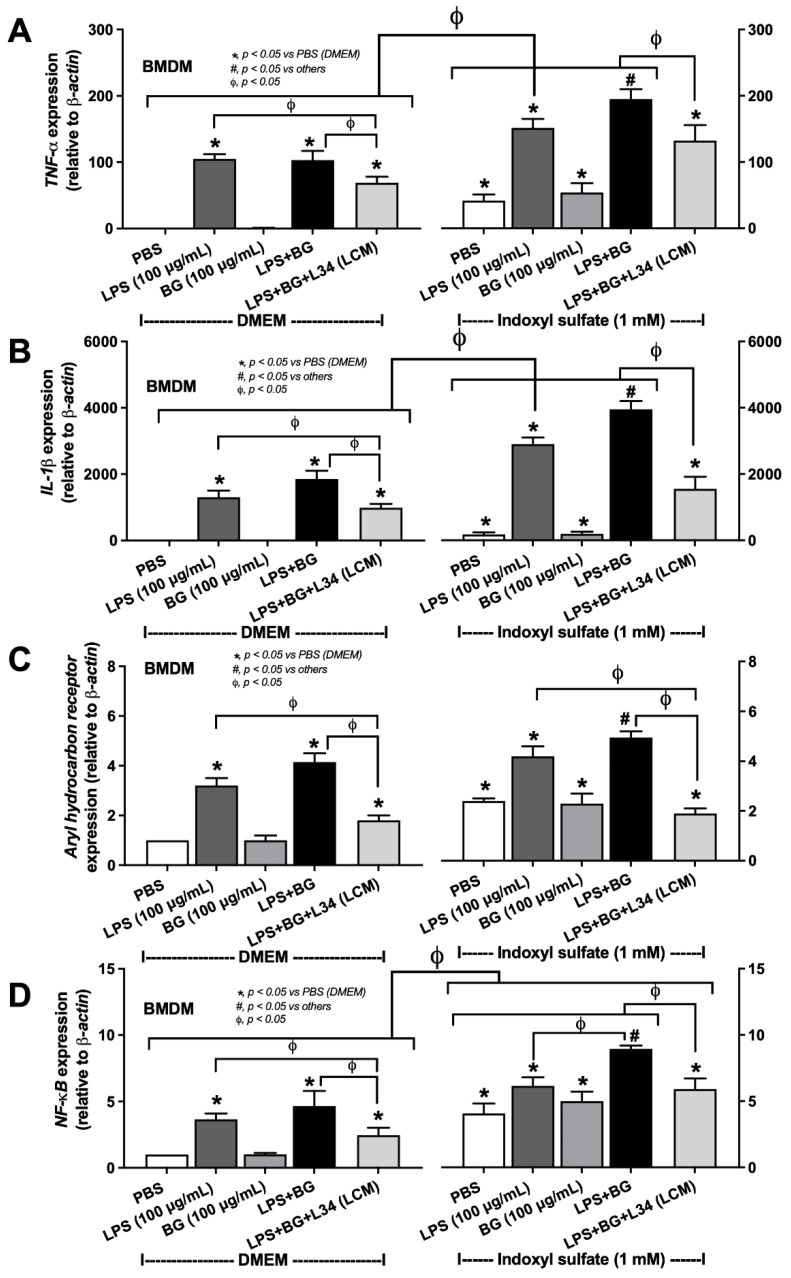
The characteristics of bone marrow-derived macrophages (BMDM) after incubation by indoxyl sulfate at 1 mM, or culture media control (using DMEM), together with lipopolysaccharide (LPS) at 100 μg/mL with or without (1➔3)-β-D-glucan (BG) at 100 μg/mL as indicated by the expression of several genes for inflammatory cytokines: (**A**) *TNF-α* and (**B**) *IL-1β*, and signaling (**C**) *aryl hydrocarbon receptor* and (**D**) *NF-κB* are demonstrated (independent triplicate experiments were performed for the in vitro experiments).

**Figure 8 ijms-23-02511-f008:**
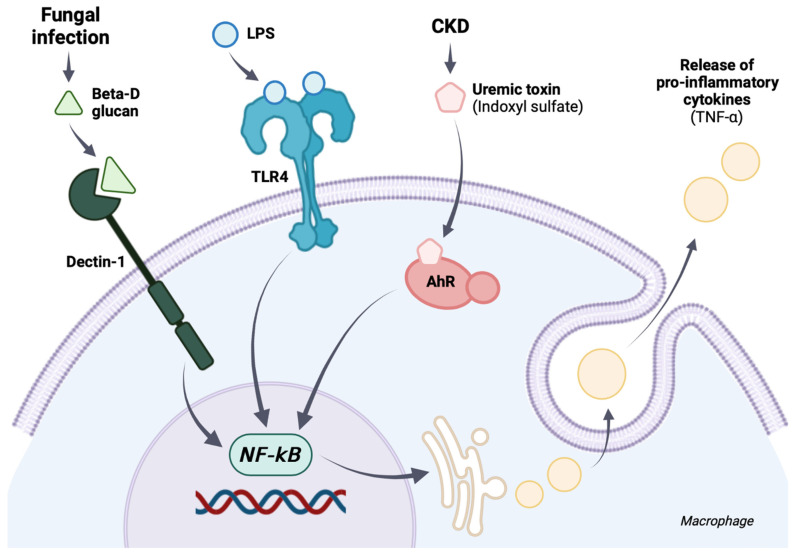
The proposed working hypothesis demonstrates the possible additional proinflammatory impact among lipopolysaccharide (LPS), (1➔3)-β-D-glucan (BG), and indoxyl sulfate through *NF-κB* transcription factor that is activated by TLR4, Dectin-1, and aryl hydrocarbon receptor.

## Data Availability

Not applicable.

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
