# Peer review of "Uremia-Induced Gut Barrier Defect in 5/6 Nephrectomized Mice Is Worsened by Candida Administration through a Synergy of Uremic Toxin, Lipopolysaccharide, and (1➔3)-β-D-Glucan, but Is Attenuated by Lacticaseibacillus rhamnosus L34"

_ijms, 2022, doi:10.3390/ijms23052511_

Round 1

Reviewer 1 Report

Tungsanga et al ahow us a very interesting study about the possible treatment of CKD with probiotics. This would be a great advantage, not only in cost but also in the aggressiveness of the treatment and its possible consequences. However, regarding the paper, I would like to indicate some comments and suggestions for changes that, in my opinion, are relevant for its acceptance.

  1. Figure 1 A-I symbols and size used in the graphs make it difficult to follow. It would be advisable to increase the size or format.
  2. In figure 1-O how authors explain that the % area of kidney fibrosis is higher in mice without candida and L34 than mice with candida but without L34?
  3. In figure 2, kidney histopathological pictures of those treated with L34 seems to show more tubular dilation than those treated only with Candida. What could this be due to?
  4. To be able to confirm that the enhancing benefits of probiotics in CKD, Have authors checked their results in another experimental model of CKD to confirm them, like the adenine-diet one? This would allow demostrating its role in the disease and not only this improvement in a specific experimemtal model.
  5. Have the authors studied what happens to the inflammasome in this model? Specifically, NLRP3, which is known to be activated in response to pathogens (among them, LPS)? It would be interesting to see how it changes with or without the administration of probiotics. LPS alone, BG alone or the combination of both have different efects on NLRP3 expression levels?

Author Response

Thank you for taking time to review out manuscript.

Please find our response to each review comment below.

Reviewer 1

Tungsanga et al show us a very interesting study about the possible treatment of CKD with probiotics. This would be a great advantage, not only in cost but also in the aggressiveness of the treatment and its possible consequences. However, regarding the paper, I would like to indicate some comments and suggestions for changes that, in my opinion, are relevant for its acceptance.

Major points

  1. Figure 1 A-I symbols and size used in the graphs make it difficult to follow. It would be advisable to increase the size or format.

Thank you for the comment. We have increased the size of fig 1 graphs and symbols accordingly.

  1. In figure 1-O how authors explain that the % area of kidney fibrosis is higher in mice without candida and L34 than mice with candida but without L34?

Although this might be due to an anti-fibrotic effect, the results were non-statistically significant and inappropriate to mention this hypothesis yet. We have added a remark in the results (page 3, line 121-123) now read: “Although there was a trend of the more severe renal fibrosis in non-Candida 5/6Nx versus Candida-5/6Nx, the value was non-statistically significant (Fig 1O).”

  1. In figure 2, kidney histopathological pictures of those treated with L34 seems to show more tubular dilation than those treated only with Candida. What could this be due to?

This might be because of the area next to the surgical resection area. We have added a remark in the results (page 3, line 123-125) now read: “Notably, the tubular dilatations in renal pathology of 5/6Nx model, despite a non-tubular ligation maneuver, might be due to the surgical scars (Fig 2).”

  1. To be able to confirm that the enhancing benefits of probiotics in CKD, Have authors checked their results in another experimental model of CKD to confirm them, like the adenine-diet one? This would allow demostrating its role in the disease and not only this improvement in a specific experimemtal model.

We thank the reviewer for this expert comment and will study the model. Here, we have added more discussion and references on the adenine-induced CKD model in the discussion (page 13, line 372-378) now read: “Probiotic tests on other CKD models might be interesting because of the possible difference in interference of gut dysbiosis and inflammatory responses in in-dividual CKD models. For example, gut dysbiosis might be more prominent in the oral adenine-induced CKD model due to its direct effect on intestinal microbiota with inflammasome-related inflammatory mechanisms (crystal-mediated inflammation) (Aranda-Rivera, A.K//. Antioxidants 2022, 11,246.// Vilaysane A,. J Am Soc Nephrol. 2010;21(10):1732-1744 // Hutton HL,. Nephrology (Carlton). 2016 Sep;21(9):736-44.// Rahman A,. PLoS One. 2018;13(2):e0192531.).”

  1. Have the authors studied what happens to the inflammasome in this model? Specifically, NLRP3, which is known to be activated in response to pathogens (among them, LPS)? It would be interesting to see how it changes with or without the administration of probiotics. LPS alone, BG alone or the combination of both have different effects on NLRP3 expression levels?

We thank the reviewer for this expert comment and will study the topic. Here, we have added more references on the correlation between inflammasome and adenine-CKD model in the discussion (page 13, line 372-378) now read: “For example, gut dysbiosis might be more prominent in the oral adenine-induced CKD model due to its direct effect on intestinal microbiota with inflammasome-related inflammatory mechanisms (crystal-mediated inflammation). With gut dysbiosis attenuation and anti-inflammatory properties of the probiotics here, further exploration in other CKD models and clinical studies for a potential application in CKD are warranted.”

Reviewer 2 Report

A very interesting manuscript with a lot of analysis done. The authors analyzed the results of their study in a thorough manner. Only some minor issues should be adressed: please use italics for microorganisms names, in vitro, etc., please use weeks instead of wks.

Author Response

Thank you for taking time to review out manuscript.

Please find our response to each review comment below.

Reviewer: 2

Minor comments:

  1. A very interesting manuscript with a lot of analysis done. The authors analyzed the results of their study in a thorough manner. Only some minor issues should be addressed: please use italics for microorganisms names, in vitro, etc., please use weeks instead of wks.

We thank the reviewer for the comments and have corrected those accordingly.

Round 2

Reviewer 1 Report

Thanks for the modifications